# Global Lipidome Profiling Revealed Multifaceted Role of Lipid Species in Hepatitis C Virus Replication, Assembly, and Host Antiviral Response

**DOI:** 10.3390/v15020464

**Published:** 2023-02-07

**Authors:** Khursheed Ul Islam, Saleem Anwar, Ayyub A. Patel, Mohammed Tarek Mirdad, Mahmoud Tarek Mirdad, Md Iqbal Azmi, Tanveer Ahmad, Zeeshan Fatima, Jawed Iqbal

**Affiliations:** 1Multidisciplinary Center for Advanced Research and Studies, Jamia Millia Islamia, Jamia Nagar, New Delhi 110025, India; 2Department of Clinical Biochemistry, College of Medicine, King Khalid University, Abha 62529, Saudi Arabia; 3College of Medicine, King Khalid University, Abha 62529, Saudi Arabia; 4Department of Medical Laboratory Sciences, College of Applied Medical Sciences, University of Bisha, Bisha 61922, Saudi Arabia; 5Amity Institute of Biotechnology, Amity University Haryana, Manesar, Gurugram 122413, India

**Keywords:** lipidome, hepatitis C virus, fatty acids, glycerolipids, glycerophospholipids, antiviral response

## Abstract

Hepatitis C virus (HCV) is a major human pathogen that requires a better understanding of its interaction with host cells. There is a close association of HCV life cycle with host lipid metabolism. Lipid droplets (LDs) have been found to be crucial organelles that support HCV replication and virion assembly. In addition to their role in replication, LDs also have protein-mediated antiviral properties that are activated during HCV infection. Studies have shown that HCV replicates well in cholesterol and sphingolipid-rich membranes, but the ways in which HCV alters host cell lipid dynamics are not yet known. In this study, we performed a kinetic study to check the enrichment of LDs at different time points of HCV infection. Based on the LD enrichment results, we selected early and later time points of HCV infection for global lipidomic study. Early infection represents the window period for HCV sensing and host immune response while later infection represents the establishment of viral RNA replication, virion assembly, and egress. We identified the dynamic profile of lipid species at early and later time points of HCV infection by global lipidomic study using mass spectrometry. At early HCV infection, phosphatidylinositol phospholipids (PIPs), lysophosphatidic acid (LPA), triacyl glycerols (TAG), phosphatidylcholine (PC), and trihexosylceramides (Hex3Cer) were observed to be enriched. Similarly, free fatty acids (FFA), phosphatidylethanolamine (PE), N-acylphosphatidylethanolamines (NAPE), and tri acylglycerols were enriched at later time points of HCV infection. Lipids enriched at early time of infection may have role in HCV sensing, viral attachment, and immune response as LPA and PIPs are important for immune response and viral attachment, respectively. Moreover, lipid species observed at later infection may contribute to HCV replication and virion assembly as PE, FFA, and triacylglycerols are known for the similar function. In conclusion, we identified lipid species that exhibited dynamic profile across early and later time points of HCV infection compared to mock cells, which could be therapeutically relevant in the design of more specific and effective anti-viral therapies.

## 1. Introduction

Hepatitis C virus is a leading cause of liver disease and is one of the main reasons for liver transplantation [1]. Severe HCV infection is associated with life threatening complications and represents a leading etiology for hepatocellular carcinoma (HCC), making it an important pathogen to study [2]. Epidemiological evidence suggests that HCV infection regulates host cellular metabolism, providing insight into the study of metabolomics. [3]. HCV is an enveloped virus belonging to the family of Flaviviridae, with a 9.6 kb positive sense ssRNA genome. The genome encodes a large polyprotein precursor of around 3000 amino acids which is cleaved by host and viral proteases into 10 different structural and non-structural proteins [4]. HCV proteins are not only involved in replication and maturation of viral progeny, but also regulate various host cellular processes. Increasing evidence suggests that HCV proteins such as NS5A perturb the lipid metabolism by inhibiting AMP-activating protein kinase (AMPK) phosphorylation [5].

Metabolic regulation of lipids takes place through important organelles called lipid droplets (LDs). These structures facilitate communication between different organelles and associate with mitochondria, peroxisomes, and endoplasmic reticulum (ER) [6]. LDs are made up of a hydrophobic core of neutral lipids which mainly consist of triacylglycerols (TAGs), cholesterol esters, a phospholipid monolayer, and a large group of proteins that facilitate the signaling interactions and help in movement of LDs within the cell. Viral infections as well as synthetic dsRNA induce LDs as early as 2 h post viral recognition in Zika and Dengue virus (DENV) [7]. HCV is also known to induce lipid droplets accumulation in cell culture as well as in vivo. HCV core and non-structural protein (NS5A) that are important for virion replication and encapsidation have been shown to co-localize with lipid droplets during HCV infection [8], thus signifying their role in viral replication and encapsidation. Interaction of diacylglycerol acyltransferase-1 (DGAT1) with HCV core and NS5A is required for trafficking of these proteins to LDs and eventually generate infectious virion particles [9]. In addition, LDs have been studied to accumulate complex immune protein networks such as viperin that regulate innate immune response against viral infections [10,11]. Loss of LD mass impairs immune response against Sendai virus (SeV) infection, which further signifies the role of lipids in innate immune response [11]

Lipids play a crucial role in different stages of viral replication cycle. Positive sense RNA viruses, including HCV, are known to utilize lipids as receptors or co-factors for viral entry in the host cells [12]. HCV utilizes lipids as building blocks for replication complexes, cellular distribution of viral proteins, assembly, and release of progeny virion particles. By influencing lipid composition of cellular membranes, HCV replicates well in cholesterol and sphingolipid-rich membranes [13]. HCV, like other RNA viruses, establishes special sites of replication on the surface of ER, called membranous web (MW); these structures are the result of membrane remodeling induced by viral and host factors [14]. As well as serving as a sophisticated niche for viral maturation, MWs also help viruses to escape host immune system. Lipid kinases, belonging to the phosphatidylinositol-4-kinase (PI4K) III family, are important factors to regulate the biogenesis of membranous web induced by HCV proteins, NS5A, and NS4B [15]. As well as providing a platform for replication and maturation of viruses, many roles have been described recently for lipids during immune response, calcium homeostasis, modulation of lipid rafts, and their interaction with pathogen recognition receptors (PRRs). Early induction of LDs corroborates with the early release of dsRNA intermediates in cytoplasm during HCV infection, indicating the onset of viral genome sensing at early viral infection [16]. Another study shows the expression of interferon response at 2 h and 8 h of vesicular stomatitis virus (VSV) infection [17]. These studies signify the importance of early viral infection in viral sensing and immune response.

Moreover, numerous lipid species have been demonstrated for their diverse function in viral propagation and host antiviral response. Lysophosphatidic acid (LPA), an important lipid species has been studied to block innate immune response against SARS-CoV-2 infection [18]. HCV nonstructural proteins including NS4B have been shown to bind PIP, cardiolipins, and phosphatidylserine (PS) which might be crucial for viral replication [19]. Similarly, NS5B specifically binds to sphingomyelin which enhances RNA polymerase activity of NS5B [20]. Phosphatidylethanolamine (PE) is known to play critical role in HSV-1 replication and assembly [21]. These studies provide strong evidence regarding involvement of lipids in different aspects of viral life cycle and host immune response.

In this study, we observed a significant enrichment of lipid LDs from early (2 h and 10 h) to later HCV infection (day 7) compared to mock cells. Using a lipidomic study, we identified an enrichment of LPA and several phosphoinositide (PIs) lipids, Hex3cer, GD1, TAG, and NAPE exclusively at early HCV infection. At day 7 post-infection, we identified an enrichment of glycerophospholipids such as PE, free fatty acids, gangliosides, and NAPE. Overall, our study identified lipids that exhibited a dynamic profile across early and later time points of HCV infection in Huh7 cells. This dynamic alteration of lipid regulation during HCV infection can be used to develop new drug targets.

## 2. Materials and Methods

### 2.1. Reagents and Antibodies

Dulbecco’s Modified Eagle Medium (DMEM), fetal bovine serum (FBS), phosphate buffer saline (PBS), 4-(2-hydroxyethyl)-1-piperazineethanesulfonic acid (HEPES, 1M), L-glutamine (100X), non-essential amino acids (100X), and antibiotics (Pen, Strep, Amphotericin B, 100X) were purchased from Thermo Fisher Scientific (Gibco^TM^, Waltham, MA, USA). All the cell culture dishes were purchased from Corning. Microscopic glass cover slips were purchased from Blue Star. Molecular grade paraformaldehyde, triton X-100, and 4,4-difluoro-1,3,5,7,8-pentamethyl-4-bora-3a,4a-diaza-s-indacene (BODIPY) were purchased from Thermo Fisher Scientific. ZORBAX RRHD Eclipse Plus C18 column with 1.8 µ particle size, 2.1 mm inner diameter, and length 100 mm were purchased from Agilent Technologies, Santa Clara, CA, USA. Anti-actin and anti-HCV core antibodies were purchased from BIO-RAD (Hercules, CA, USA) and GeneTex (Irvine, CA, USA) respectively. Anti-dsRNA antibody was purchased from Millipore (Merk-Sigma, Burlington, MA, USA).

### 2.2. Cell Lines and HCV Cell Culture

Human hepatoma cell lines Huh7 and subline Huh7.5 were obtained from Dr. Ranjith Kumar, University School of Bio-Technology, Guru Gobind Singh Indraprastha University, New Delhi, India. Huh-7 and Huh7.5 cells were maintained in DMEM supplemented with 10% FBS, 1% pen-strep antibiotic cocktail, 1% HEPES, and 1% non-essential amino acids. Cells were incubated in 5% CO_2_ at 37 °C.

Since JFH-1 is known to propagate well in subline Huh7.5 owing to its defective immune response induced through point mutation in retinoic acid inducible gene-I (RIG-I) [22], these cells were utilized for generation of HCV particles while all other experiments were conducted in Huh7 cells. The J6/JFH-1 (genotype 2a) RNA was in vitro transcribed and electroporated into Huh7.5 cells using Gene Pulser Xcell Total system (BioRad). Cell free virus contained in the cell culture supernatant was also propagated in Huh7.5 cells as described previously [23]. The viral titer was expressed as focus forming unit (ffu) mL^−1^, which was observed by the average number of HCV-core-positive foci detected at the highest dilutions as described earlier [23]. Cells treated with cell culture supernatant collected from Huh7.5 cells expressing JFH-1/GND (replication defective virus) were used as a negative control and are represented as mock cells. Huh7 cells were infected with HCV (MOI ~0.5) for further experiments.

Huh7 cells were incubated with J6/JFH-1 or JFH1/GND supernatant for 4–6 h and then replaced with fresh medium. Cells were routinely maintained in complete DMEM media and passaged regularly at 70–80% confluency. Cell morphology was also checked regularly, and media was changed at an interval of 2–3 days. Mock cells were grown along with HCV infected cells to maintain every possible condition for experiments.

### 2.3. SDS-PAGE and Western Blot Analysis

Total cellular protein lysates were prepared from mock- and HCV-infected Huh7 cells at different time point of infection. Briefly, after aspirating media, cells were washed by cold 1X PBS, thoroughly scrapped, and centrifuged at 13,000× *g* for 5 min. After removing supernatant, cells were lysed using RIPA (Radioimmunoprecipitation assay buffer) lysis buffer (50 mM Tris, pH 7.5, 150 mM NaCl, 1% NP-40, 0.5% sodium deoxycholate, 0.1% SDS, 1% protease inhibitor cocktail) and kept on ice for 30 min. Further, lysed cells were centrifuged at 13,000× *g* at 4 °C for 15 min, supernatants were collected and subjected to protein estimation using Bradford reagent (Bio-Rad) [24]). Equal amount of protein was loaded on 12% SDS-PAGE gel, followed by transferring to PVDF membrane and incubated with blocking solution (5% skimmed milk in TBST) for 1 h. Further membrane was washed and incubated with anti-actin and anti-core primary antibody followed by incubation with horseradish peroxidase-conjugated secondary antibody for 1 h as shown earlier [25]. Antigen–antibody complex was detected by enhanced chemiluminescence.

### 2.4. Immunofluorescence Microscopy

Mock and HCV-infected cells were grown on glass coverslips (18 mm) at different time points of infection (2 h to day 7) and fixed with 4% paraformaldehyde and permeabilized with 0.1% triton X-100 in 1% PBS. Cover slips were incubated in primary antibody prepared in 1%BSA in 1x PBS followed by secondary antibody incubation for 1 h. Cells were immunostained using anti-dsRNA and HCV anti-core primary antibody followed by Alexa Flour-594 conjugated goat-anti-mouse secondary antibody. Lipid droplet staining was performed using BODIPY dye as described previously [26]. After staining nucleus with 4′,6-diamidino-2-phenylindole (DAPI), cover slips were mounted on glass slides and observed under Zeiss Axio observer 7 immunofluorescence microscope.

### 2.5. Lipid Extraction

Mock and HCV-infected Huh7 cells (2 h, 10 h, and day 7) were used for lipid extraction study. Three biological replicates were taken for each group for lipid extraction. Lipid extraction was done according to the previously described Folch method [27]. Briefly, cells were washed three times with ice cold PBS and scrapped gently by cell scrapper. Cells were centrifuged at 13,000× *g* rpm at 4 °C in ice cold PBS. After removing supernatant, cell pellet was further homogenized in aqueous solution for 3 min and suspended in chloroform and methanol in ratio of (1:2). Cells were shaken well and centrifuged at 5000× *g* at 4 °C for 15 min. Supernatant was transferred to another glass vial and then remaining chloroform was added and filtered through Whatman No. 1 filter paper. The extract was then washed with 0.88% KCl to remove the non-lipid contamination. The lower dense layer of chloroform containing lipid was taken by glass Pasteur pipette in 5 mL glass vial with Teflon capping. The extraction solvents were evaporated using N_2_ gas flux and stored at −20 °C until further use.

### 2.6. Ultraperformance Liquid Chromatography-Electrospray Ionization-Mass Spectrometry (UPLC-ESI-MS)

To perform reverse-phase ultrahigh-pressure liquid chromatography (UHPLC, Exion LC Sciex, Framingham, MA, USA) using Kinetex C18, 2.1 × 50 mm column (Phenomenex, Torrance, CA, USA) with a particle size of 1.7 mm column coupled to a hybrid triple quadrupole/linear ion trap mass spectrometer (4500 Q-TRAP, SCIEX, Foster, CA, USA). The dried sample was reconstituted in mobile phase solvent B before use. The sample was introduced using an autosampler with 5 mL of sample injection volume at a flow rate of 0.1 mL/min. The elution was done for 30 min, using mobile phase A and B, solvent A has 10% methanol and 90% buffer of ammonium acetate (5 mM) dissolved in water and acetonitrile (95:5) and solvent B has 5% isopropanol, 10% methanol, and 85% acetonitrile. The source temperature was 120 °C, the desolvation temperature was 350 °C, and the cone voltage was set at 40 volts (V). Capillary voltage (i.e., spray voltage) was set to 3.50 kilo volts (kV). The data were recorded in the mass range *m*/*z* 200–2000 Da in electrospray ionization (ESI) positive and negative ion mode. The data processing was done with the Analyst software, where each chromatogram was smoothened and the background was subtracted. Triplicate LC-ESI-MS datasets were acquired and the extent of reproducibility of the data was verified. Further analysis was carried out with the reproducible LC-ESI-MS data and only those peak *m*/*z* values in the mass spectra have been considered for interpretation, whose intensities are higher than 5000 (namely, threshold value to ignore the peaks with poor signal-to-noise ratios). Analysis of the acquired data was performed by two different software viz. LipidView and mass spectrometry-based lipid (ome) Analyzer and molecular platform (MS-LAMP) software. MS-LAMP software is a graphical user interface (GUI) standalone program built using Perl::Tk, based on lipid metabolites and pathways strategy consortium (LIPID MAPS; www.lipidmaps.org) [28]. Further, it needs to be noted that the data herein were analyzed qualitatively only. All the experiments were performed in biological triplicates to ensure reproducibility and accuracy.

Data were processed uploaded into MetaboAnalyst 5.0 (http://www.metaboanalyst.ca) (accessed on 8 October 2022) and normalized for uni and multivariate statistical analysis in both positive as well as negative mode separately. The initial exploratory data analysis was done by univariate one way analysis of variance (ANOVA). Unsupervised PCA method was applied thorough prcomp package and calculation was based on singular value decomposition to explain the variance in data set. Then, the PLS supervised method using multivariate regression technique was applied using plsr function provided by rpls package. The classification and cross validation were done using their corresponding wrapper function through caret package. The sparse PLS-DA (sPLD-DA) algorithm was used to effectively reduce the number of variables. The top 50 significant features were identified from ANOVA and post-hoc analysis represented in the Appendix A.

## 3. Results

### 3.1. Lipid Droplet Enrichment during HCV Infection in Huh7 Cells

HCV utilizes the host ribosomal machinery for effective expression of its structural and non-structural proteins [29]. Here, Huh7 cells were infected with HCV (MOI ~0.5) and grown for different time points of infection. The cellular protein lysates were immunoblotted using HCV core antibody and results show the significant expression of HCV core protein at different time points of viral infections (Figure 1a). HCV titer was calculated by focus forming unit assay using HCV core antibody as shown earlier [30]. To confirm the early HCV infection, we performed immunostaining using anti-dsRNA antibody at 2 h of HCV infection and results show the substantial amount of dsRNA staining (Figure 1b) as shown earlier [16].

Lipid droplets have been extensively studied to co-localize with special membrane structures at endoplasmic reticular membranes as platforms of HCV replication [31]. In addition, they have been studied to harbor antiviral proteins such as Viperin and lipid droplet associated proteins such as perilipin 2 [26], thus acting as a platform for innate immune response. To determine the enrichment of LDs during HCV infection, Huh 7 cells were infected at various time points (2 h, 4 h, 8 h, 10 h, 12 h, day 1, day 3, day 5, and day 7) and stained with BODIPY (marker for LD) as described earlier [32]. The results depict the density of LDs with respect to time of infection, where LDs were stained and visualized at 488 nm (green color). However, anti-HCV core antibody staining was used for viral infection and colocalization with LDs which appeared at day 3 infection onwards and is shown in red (595 nm) (Figure 2). This early enrichment of LDs compared to mock cells indicate their contribution in innate antiviral response which is consistent with the previous studies [11]. At days 3, 5, and 7, a significant colocalization of HCV core with LDs (yellow dots) was observed which indicates that core and LDs are exploited for virion assembly, as reported by others [33]. Collectively, these results show the significant enrichment of LDs from early to later HCV infection, suggesting that LDs are probably utilized by the virus for its proliferation; however, the host might be utilizing it for antiviral response to curb the viral infection.

### 3.2. Qualitative Analysis and Differences of Total Lipids during HCV Infection in Huh7 Cells

Since LDs are known to be crucial for the HCV replication and assembly, we sought to determine which lipids species are critically contributing to HCV life cycle. It is well studied that HCV genome is sensed at very early stage of viral infection by TLRs in endosome and by retinoic acid-inducible gene-I (RIG-I) in the cytoplasm, which triggers a cascade of reaction to stimulate antiviral response [34]. In contrast, enhanced HCV replication and assembly establishment has been shown at later time point of HCV infection [16]. Moreover, several lipids are crucial and contribute to innate response, HCV replication, and assembly processes [35]. To determine the contribution of lipid species in host innate response and HCV proliferation, we selected 2 h and 10 h for early infection when viral genome sensing and innate response occur and day 7 for later infection when viral replication and its virion assembly occur. To obtain a comprehensive kinetic view of lipid alteration at different time points of HCV infection, we performed reverse phase UP-LC-ESI-MS based lipidomic analysis of HCV infected Huh7 cells (2 h, 10 h, and day 7) and compared the results with mock cells. Statistical methods were utilized to differentiate between the lipid compositions of mock and HCV-infected cells. Through one-way analysis of variance (ANOVA), the top 50 features were identified each from positive ion mode and negative ion mode in all the four samples with *p*-value less than 0.05 (shown in Figure 3a,b and Appendix A). To demonstrate whether metabolite fingerprint differed in mock and HCV-infected samples at various time points of HCV infection, multivariate analysis was carried out. Non supervised principal component analysis (PCA) (Figure 3c,d) shows the clear clustering of HCV infected samples compared to mock samples. According to the PCA model, the total variance of data in positive ion mode was 73.5% with 48.7% contributed by PC1 and 24.8% contributed by PC2 and an additional 22.7% contributed by PC3 (18.5%), PC4 (2.4%) and PC5 (1.8%) (Appendix A). Similarly, in negative ion mode, total variance in data was 76.6% with 43.4% contributed by PC1 and 33.2% contributed by PC2 and an additional 22.9% contributed by PC3 (22.4%) (Appendix A). All the samples clustered in distinct groups in positive and negative ion mode when PC1 and PC2 with highest variance was plotted both indicating the lipidomic differentiation between mock and HCV-infected samples at different time points of infection (Figure 3c,d).

### 3.3. Differential Lipid Content in Mock and HCV Infected Huh7 Cells Using sPLS-DA

sPLS-DA (Sparse partial least squares-discriminant analysis) is a sparse version of partial least squares-discriminant analysis (PLS-DA) for clustering and classification. PLS-DA performs well for regression analysis, while for classification, the sparse version of PLS-DA (sPLS-DA) is preferred [36]. We used both PLS-DA as shown in Appendix A and sPLS-DA (Figure 4) for clustering and classification of lipid data. sPLS-DA differentiates the variables on the basis of their distinct features. We utilized this method to differentiate the lipid species present in mock and HCV-infected Huh7 cells at different time points of HCV infection (2 h, 10 h, and day 7). This method facilitated us investigating the effect of HCV infection on lipidomic profile of cultured Huh7 cells. The results show the sPLS-DA score plot between principal component 1 and 2 having highest variance, in positive and negative ion mode, respectively (Figure 4a,b). The lipid species profile of mock and HCV-infected Huh7 cells was clearly classified according to first two principal components with a cumulative contribution rate equal to 72.2% and 75.6% in positive and negative ion mode, respectively. Loading scores of lipids contributing to the differentiation of mock and HCV infected Huh7 cell culture samples were estimated, both in positive as well as negative ion mode. The lipid species contributing to the classification of control and HCV-infected samples were identified using loading scores as depicted in Figure 4c,d. Lipids having higher loading scores are considered to be relatively significant in contributing to the separation between the various groups, those with lesser loading scores are considered as less influential.

#### 3.3.1. Hierarchical Clustering Analysis of Lipid Species at Different Time Point of HCV Infection

Based on statistical analysis, more than 100 features each from positive ion mode and negative ion mode were selected. The distinct features were sufficient enough between mock and HCV infection at 2 h, 10 h, and day 7. To further understand the features, a hierarchical clustering analysis was performed on the basis of degree of similarity between the lipid abundance profiles within the samples. A heat map was generated to identify distinct features among each group under study as described earlier [37]. We observed dynamic change in lipid species profile at selected time points of HCV infection. The variation of different ion features in the form of a heat map in positive ion mode and negative ion mode is shown in Figure 5. To further simplify the heat map, we provided arbitrary naming to each group as depicted (Lp1–Lp11 and Ln1–Ln6) in Figure 5 and Appendix A.

In the first group (Lp1), lipid species were not enriched at any time points of HCV infection, compared to mock cells, thus highlighting the role of these lipids (PIP:33;3, GD3 36:2;2, PIP2 34:0, and GD3 46:1;2) in normal cellular metabolism. Lp2 consists of lipids (GT3 32:0;3, PIP 45:6, GM2 38:2;2, and NAPE 48:4+NH4) which are exclusively upregulated at day 7 infection besides showing a little expression in the mock cells. Based on their upregulation at later time point of HCV infection, these lipids species seem to assist HCV replication and virion assembly. Their downregulation during early infection may be the host strategy to block their synthesis for the benefit of viral propagation. Lp3 shows the expression of certain lipid species exclusively at day 7, among which PE 32:4 and FFA 14:1 are known to contribute in viral replication and assembly events. Tomato bushy stunt virus (TBSV) utilizes PE for its genome replication [38]. We observed enhanced free fatty acid (FFA) expression at later HCV infection which is in agreement with earlier report [39]. Some of the lipid species, as depicted by Lp4, were upregulated at early as well as later HCV infection which might have multiple roles in HCV life cycle as GD3 36:2;3, CDPDAG 36:6, and Hex3cer 30:0;2 have previously been shown to affect viral infection. In addition to this, we observed upregulation of ceramide, such as Hex3cer 30:0:2 which inhibits viral entry [40]. Moreover, GD3 gangliosides enhance apoptosis potentiated by blocking nuclear translocation of NF-κB [41]. Therefore, their upregulation at 2 h of HCV infection signifies a potential role in preventing innate immune response against viral infection. As we observed PIs upregulation at early and later HCV infection, previous study also signifies their role in HCV infection, replication, and egress [42]. As depicted by Lp5, certain lipids (PIPs, PG, PC, and GMs) were found to be strongly upregulated at 2 h infection. PIs and phosphotidylcholine (PCs) are the membrane lipids and their perturbation might be a viral strategy for fusion and entry events, as studied previously [43]. We observed an upregulation of glycerophospholipids (PG) at early HCV infection which may be as a result of their involvement in innate immune response because these lipids regulate IFN response against influenza A virus (IAV) [44]. Lp6 show the expression of PIP, PC, and GM lipids enriched at 2 h infection; however, their expression decreased at later infection. GM1 gangliosides assists in viral entry and induce membrane curvature during infection of Simian Virus 40 (SV40) [45]. In LP7, we observed almost equal distribution of phosphoinositides, free fatty acids, glycosphingolipids, etc. at every selected time points of HCV infection while their expression was comparatively less in mock cells. LP8 consists of lipid species enriched at 2 h and 10 h, these lipid species might be expressed either as an antiviral strategy by host or virus may exploit its biosynthetic pathways for efficient viral entry in the cells. Among these lipids, LPA has previously been shown to block antiviral response against SARS-CoV-2 infection [18]. LP9 shows the enrichment of specific lipids (GD1 46:2;2, GM1 36:2;2, GM2 44:1;2, GD1 46:0;2, Hex3Cer 30:1;2) at 10 h and day 7, which provides a clue regarding their role in viral attachment, viral packaging, and egress because their concentration remains consistent to day 7 which is the time for maturation and release of virus particles. Glycolipids such as GM1 have been studied to assist in entry of SARS-CoV-2 virus [46]. These lipids make up the special plasma membrane regions called lipid rafts that have been previously shown to assist in HCV replication [47]. Lp10 and LP11 depict the strong enrichment of various lipids including GD3 36:0;2, NAPE 40:4+NH4, TAG 56:6+NH4, and GM1 32:1;2, etc. at 10 h of HCV infection compared to mock. These lipids might be assisting in innate immune response which needs further studies.

Lipid species which were not detected in positive ion mode might be leached and detected in negative ion mode as shown in Appendix A. The clusters were labeled Ln1 to Ln6. Ln1 shows the upregulation of lipid species at day 7. Ln2 represents the enhanced enrichment of lipid species, which might be contributing to the cell’s normal function. Ln3 and Ln4 show the enrichment of lipid species at 2 h of HCV infection and in mock cells, respectively. Ln5 shows the enrichment of lipids at 10 h of HCV infection. Ln6 shows the lipid species that were enriched at 2 h and 10 h of HCV infection including its enrichment in mock cells as well. Collectively, these lipid species enrichment in mock cells as well as at various time point of HCV infection indicate their role at different stages of viral propagation and involvement in host innate antiviral response which needs more in-depth study.

#### 3.3.2. Lipidomic Analysis Reveals Major Lipid Classes Variation at Different Time Points of HCV-Infection

Different lipid species were categorized in their respective lipid classes by MS-LAMP general lipidome software. The variation in different lipid classes with respect to early and later HCV infection is shown in Figure 6a,b. The lipid classes that were significantly varied among mock and HCV infected cell culture samples include fatty acyls, glycerolipids, glycerophospholipids, prenol lipids, saccherolipids, and sterol lipids. Glycerophospholipids are the main components of lipid droplets, which are the endoplasmic-derived organelles that store neutral lipids. Lipid droplets are extensively studied for viral assembly during HCV infection. However, proteomic and lipidomic analysis using mass spectrometry experiments have revealed different other putative roles of this organelle during viral infection [31]. The results show an increased level of glycerophospholipids at early time point of HCV infection (10 h in positive ion mode and 2 h and 10 h in negative ion mode), thus reveal the importance of this class of lipids during the onset of innate immune response which begins in the few hours of HCV post infection [48]. Glycerophospholipids make a prime contribution in the biosynthesis of lipid droplets which are now being explored as the platform for innate immune response against viral infections [11]. Our study provides insights regarding the involvement of this class of lipids in innate immune response at early HCV infection (Figure 6a,b). Similarly, glycerophospholipids and glycerolipids were found to be enriched at early time points of HCV-infection. Synthesis of these lipids is carried out by a mitochondrial enzyme glycerol-3-phosphate acyltransferase (GPAM), which prefers saturated fats as its substrate. Our study indicates the increased level of glycerolipids at early HCV infection which might be due to over-expression of enzyme glycerol-3-phosphate acyltransferase that remains to be elucidated. We also observed significant level of fatty acyles at 10 h and 7 days HCV infection as fatty acid synthesis, which was previously observed to be important for HCV propagation in the host cell. Our observation is further supported by a previous study where fatty acid synthesis was found to be critical for HCV replication [49].

## 4. Discussion

Chronic hepatitis C is a global health concern, affecting 3% of world’s population [49]. Since there is no vaccine available for this disease, antivirals targeting viral proteins remains the only standard of care. However, targeting viral proteins is limited in terms of drug-induced mutations which increases the chance of antiviral resistance [50]. In light of this, it is crucial to identify new drug targets. The biosynthetic pathways of important lipids can be used as potential drug target for treatment, avoiding the risk of drug induced mutations [51]. HCV has been shown to alter the lipid content of infected cells, potentially promoting viral replication by compromising the host’s antiviral response. This is supported by evidence that HCV infection leads to a decrease in serum cholesterol and low-density lipoprotein cholesterol in patients. This alteration in lipid levels can be reversed upon successful viral eradication through antiviral treatment [52]. Additionally, while HCV has been extensively studied for its ability to exploit cellular lipids for its proliferation, its involvement in host immune response is still not fully understood. The alteration of lipids during HCV infection is well-established, but a more detailed understanding of the involvement of different lipid species is needed through a comparative analysis at early and later stages of infection. Therefore, our study aimed to qualitatively evaluate the differences in the total lipidome and lipid droplet content of mock and HCV-infected cells at different time points of HCV infection. The goal was to gain insight into the critical involvement of lipid species in innate immune response and inflammatory response at early stages of HCV infection, which marks the beginning of viral genome sensing and innate immune response [53].

Here, we performed a kinetic study to check the lipid droplet density at early and later time points of HCV infection. A significant variation in lipid droplet density with respect to time of HCV infection was observed (Figure 2). Our study revealed the perturbation of various lipid species belonging to various lipid classes at early (2 h and 10 h) and later (day 7) time points of HCV infection in Huh7 cells as summarized in Figure 7. PI- and PC-based lipid species were considerably enriched at selected time points of HCV infection. PIs are regulated by PI3-kinases pathway and this pathway is exploited by HCV NS5A to modulate cellular lipids, as reported earlier [54]. Their involvement in viral entry, replication, and assembly is also reported [24]. We observed enhanced level of PIPs in early time point of HCV infection, suggesting their role in viral binding to host cells as PI3-kinases are activated on viral binding, as reported earlier [55]. PIPs can direct cell signaling in a pro-viral manner and can directly bind to viral proteins during HCV infection [56]. Apart from viral binding, PI(4)P plays a pivotal role in innate immune response as they can bind to NLRP3 at trans Golgi networks [57]. Another important phospholipid, PE was identified at later HCV infection which has previously shown to assist viral replication, entry, and budding. Thus, targeting their biosynthetic pathways could be an important strategy to block viral replication. We also identified enhanced expression of glycerophospholipids and glycerolipids at early HCV infection (Figure 6). Glycerophospholipids are the important components of lipid droplets while glycerolipids represent an abundant lipid class playing versatile role in energy storage and membrane structure. The enzymes that regulate glycerolipid levels in the cell also modulate lipid intermediates and serve as signaling nodes influencing a wide range of physiological processes [58,59]. Thus, their alteration after HCV infection provides insights in exploring this class of lipids thoroughly with respect to viral infection. Our study further reflected the decrease in sphingolipids at each time points of HCV infection compared to mock cells which is consistent with the previous report showing the decrease in sphingomylin in JFH-1-infected Huh7 cells [60] which may either be the viral or host strategy that needs an in-depth study. We also observed many glycosphingolipids (GMs) at later HCV infection. These lipids make up the special membrane structures called lipid rafts which provide a platform for viral maturation as reported before [61]. Another important lipid species, LPA which was identified at early HCV infection in our study has recently been studied to block IRF3-mediated interferon response via Rho-associated kinases (ROCK kinases) in SARS-CoV-2 infection [18]. Its expression at early hours of infection might be the viral strategy to block innate immune response which needs further investigation.

Lipidomic study of HCV infected cells at early and later infection may reveal a novel drug against HCV as this virus exploits host lipidome for its efficient propagation inside the host. Manipulating the host targets, important for viral life cycle, is gaining importance as they possess a potential for broad spectrum activity against multiple viruses belonging to same genus or family. In addition to this, such strategy may have high barriers to drug resistance compared to virus-targeted agents [62]. Our study identified lipid species perturbed at different time points of infection, specific to HCV genome sensing, replication and assembly. This provides a clue regarding inhibiting discrete steps of viral life cycle by targeting pathways/enzymes that regulate synthesis of such specific lipids or their precursors. Collectively, this global lipidome study has revealed numerous lipid species at different time point of HCV infection which has opened new avenues for further in-depth studies. We have identified many lipid species; however, their fate in the HCV life cycle is not known yet. In contrast, we found several lipids which are important for proliferation of many viruses including HCV. These lipids or their pathways can be utilized for therapeutic purposes.

## Figures and Tables

**Figure 1 viruses-15-00464-f001:**
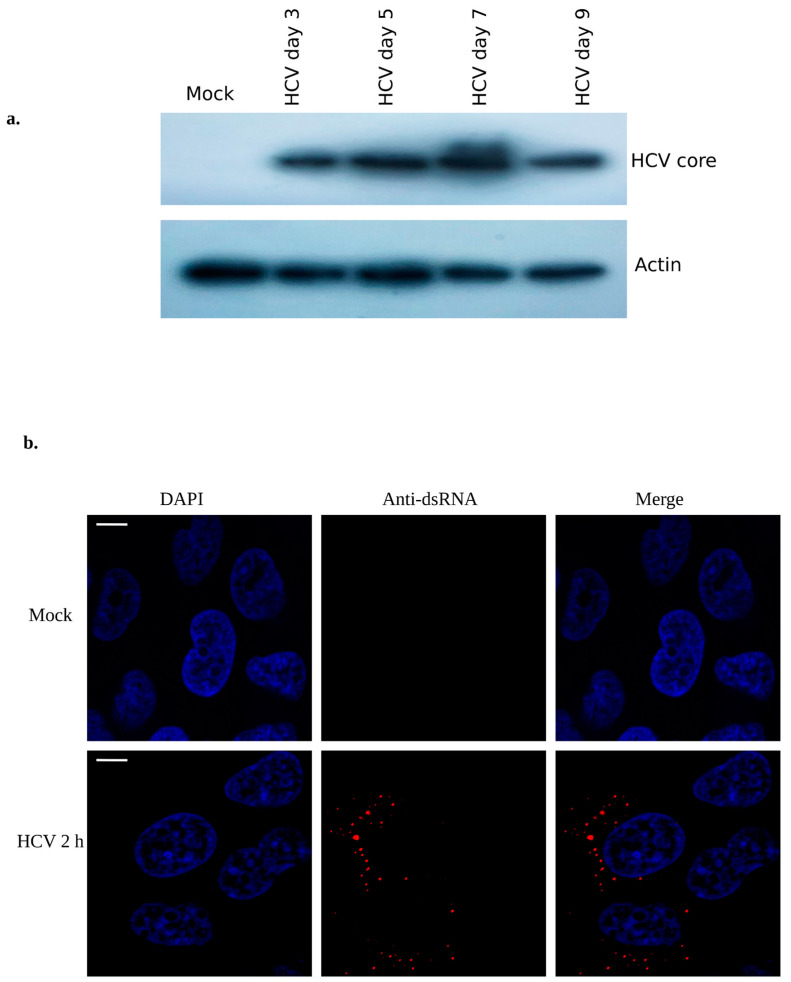
Determining the HCV infection: (**a**) Mock (Huh7) cells were infected with ~0.5 MOI of HCV. Protein lysates were immunoblotted using HCV core antibody. Actin antibody was used as input control for protein loading. (**b**) Mock (Huh7) and HCV infected cells (~0.5 MOI) were grown on sterilized cover slip for 2 h and immunostained using anti-dsRNA primary antibody followed by secondary antibody (goat anti-mouse; 595 nm) as described in the method section. The image was captured using immunofluorescence microscopy (Zeiss Axio observer 7), with scale bar of 10 μm.

**Figure 2 viruses-15-00464-f002:**
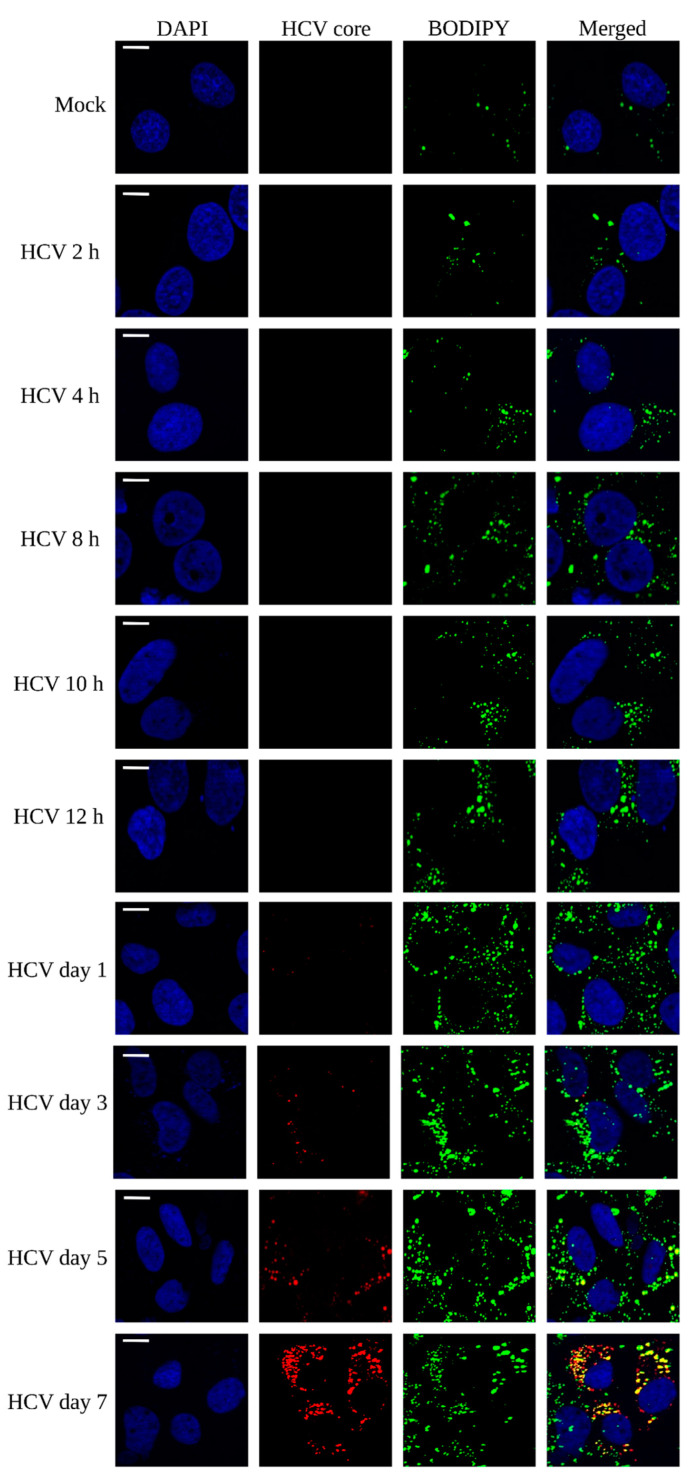
Lipid droplets staining at different time points of HCV infection: Mock and HCV infected Huh7 cells were grown for 2 h, 4 h, 8 h, 10 h, 12 h, 1 day, 3 days, 5 days, 7 days, and stained using BODYPI dye for LD enrichment. HCV core was immunostained using anti-HCV core antibody for HCV infection. LD staining was measured at 488 nm (shown in green), and HCV core was measured at 595 nm (shown in red). DAPI was used for staining of nucleus (shown in blue). Scale bar: 20 μm.

**Figure 3 viruses-15-00464-f003:**
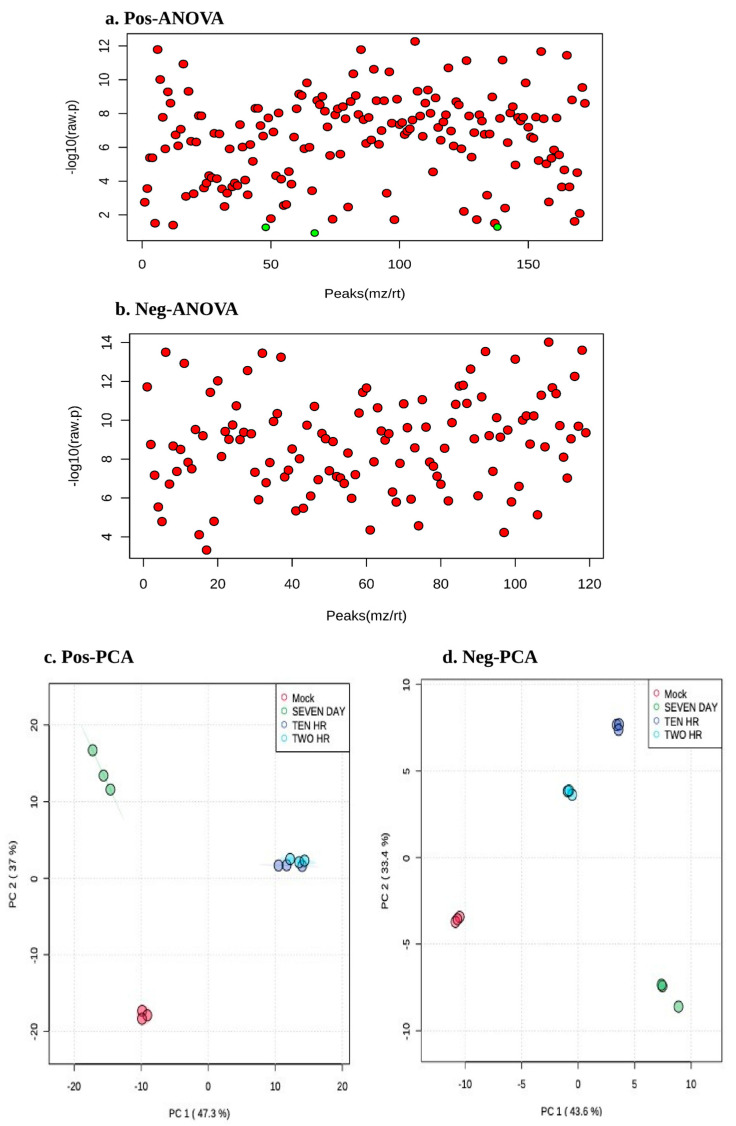
Qualitative analysis of total lipid species during HCV infection: Mock and HCV-infected Huh7 cells (2 h, 10 h, and day 7) were used for total lipid content. Panels (**a**,**b**) show the important features selected on the basis of one-way ANOVA with a *p*-value threshold = 0.05 in positive and negative ion mode respectively. Panels (**c**,**d**) show the contribution of principal component PC1 and PC2 in positive and negative ion mode, along with their %age of variance.

**Figure 4 viruses-15-00464-f004:**
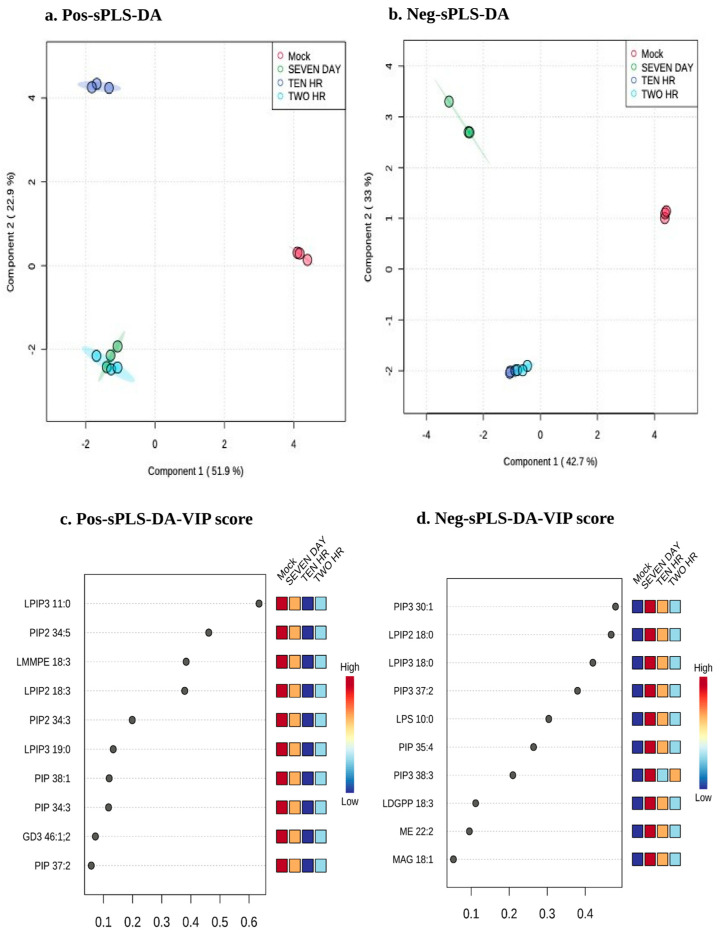
Differential lipid species identified during HCV infection: Mock and HCV-infected Huh7 cells (2 h, 10 h, and day 7) were used for differential analysis of lipid species where panel (**a**,**b**) shows PLS-DA score plot between selected PCs from positive and negative ion mode, respectively. (**c**,**d**) Features identified by sPLS-DA model for a given component. Variables are ranked by the absolute values of their loading scores. The colored boxes on the right indicate the relative concentration of metabolite under study. The red color indicates highest concentration and blue indicates the lowest concentration of the metabolite under this study.

**Figure 5 viruses-15-00464-f005:**
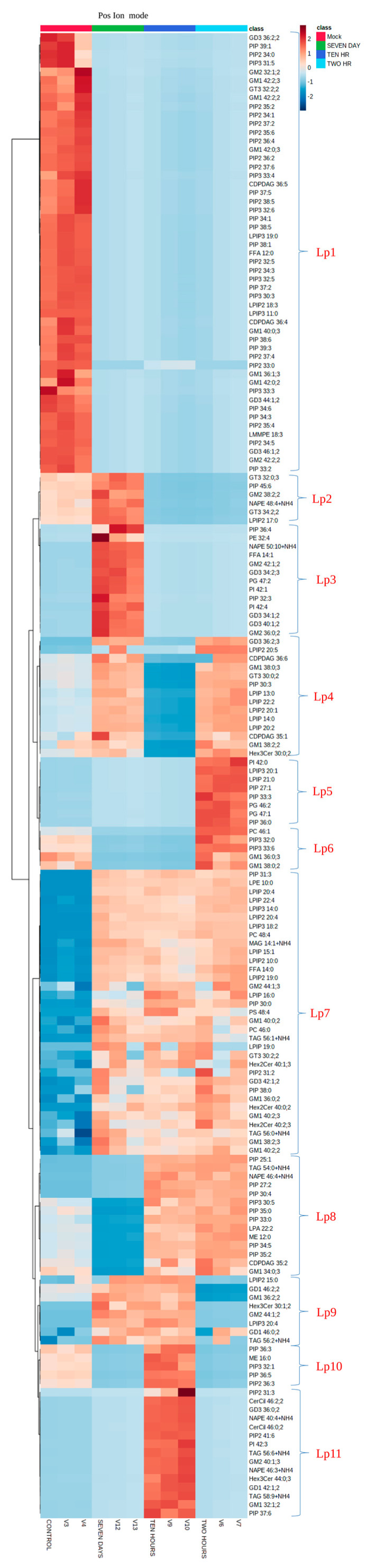
Clustering analysis for lipid species during HCV infection: Mock and HCV-infected Huh7 cells (2 h, 10 h, and 7 days) were used for clustering analysis and results show in the form of heat map, distance measure using Euclidean, and clustering algorithm using ward D. Each colored cell indicates the peak area value of corresponding lipid species, cardinal red color indicates high abundance, and blue color indicates low abundance of the lipid molecule. Red, green, blue, and cyan indicate different samples under study. The heat map shown here is for positive ion mode.

**Figure 6 viruses-15-00464-f006:**
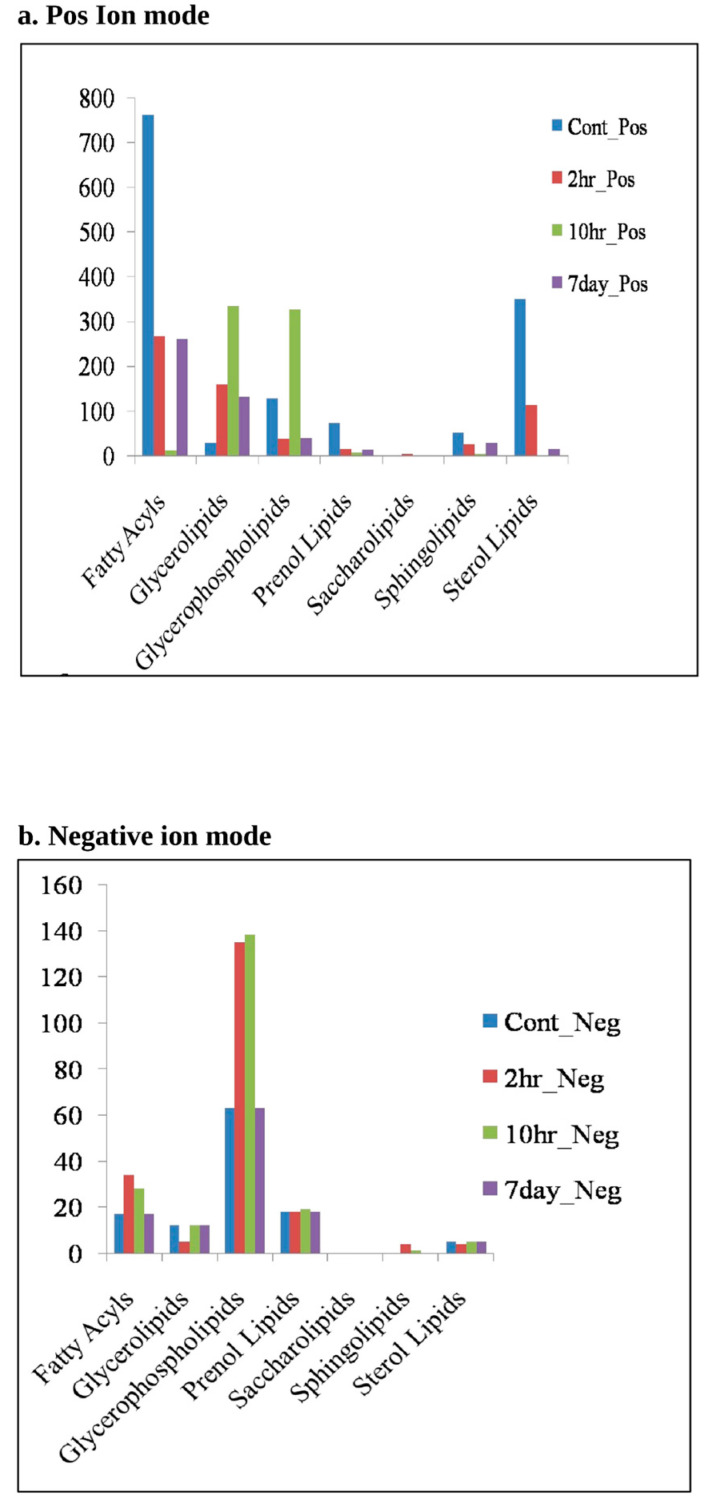
Lipidomic study in mock vs. HCV infected cells: Figure shows the overall distribution of lipid classes among different samples. (**a**) Panel shows the distribution of lipid classes in positive ion mode while (**b**) shows distribution of lipid classes in negative ion mode among mock, and HCV infected cells at 2 h, 10 h, and 7 days.

**Figure 7 viruses-15-00464-f007:**
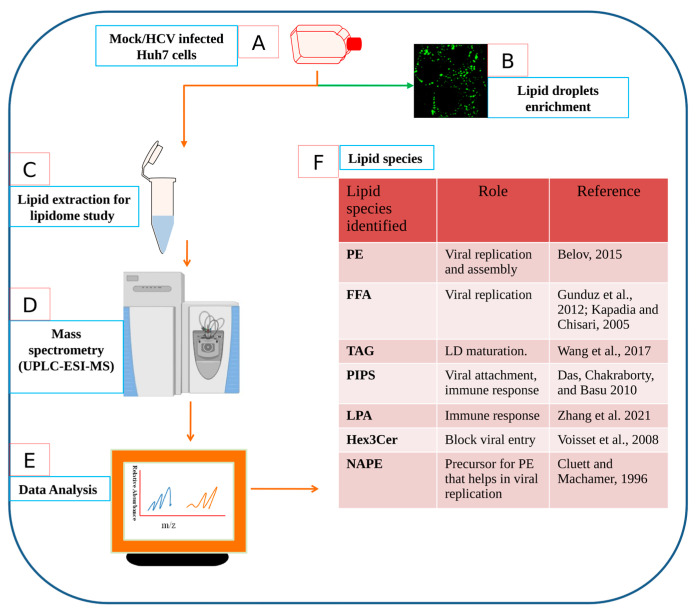
Putative model illustrates the host lipidome exploited by HCV. Host lipidome is associated with HCV life cycle. To test this hypothesis, we performed time kinetics of lipid droplet density using BODIPY dye and visualized by immunofluorescence microscopy. We observed significant enrichment of LDs at early and later time points of HCV infection. To further investigate the important role of lipid species in HCV proliferation and host antiviral response, we have performed lipidome study using UPLC-ESI-MS. This putative model provides a concise summary of workflow of the study. (**A**) Mock and HCV infected Huh7cells were grown and utilized for (**B**) lipid droplets (LDs) staining using BODIPY dye (Figure 2) and (**C**) lipids extraction using Folich method which is given in detail in material and method section. Further, (**D**) reverse-phase ultrahigh-pressure liquid chromatography (UHPLC, Exion LC Sciex, USA) was used to perform lipidomic study of given samples (Mock, 2 h, 10 h, day 7) as shown in Figure 5. (**E**) Analysis of the acquired data was performed by two different software viz. LipidView and mass spectrometry-based lipid (ome) Analyzer and molecular platform (MS-LAMP) software. (**F**) Some of the important lipid species identified significantly in our study which are listed here with their potential functions. Among these lipids identified in our study, PE has previously been studied to positively regulate viral replication [63] while as FFA enrichment has been shown to cause ER stress [64] which in turn blocks antiviral response thus helps in efficient viral replication [48]. Upregulation of phosphoinositides (PIPs) were also observed in our study which have previously been shown to assist replication [19], viral entry [55] and inflammatory response [57]. Moreover, our study identified enrichment of LPA and Hex3cer lipids which are known to block innate immune response [18] and viral entry [40]. An unusual lipid molecule (NAPE) identified in our study was previously shown to associate with Vaccinia virus [65]. Hijacking of host lipid synthesis to meet the increased demand of viral replication and other aspects of its survival is the strategy of viruses to exploit host metabolism. Thus, making this process an important target for new antivirals.

## Data Availability

Not applicable.

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
