# Peer review of "Global Lipidome Profiling Revealed Multifaceted Role of Lipid Species in Hepatitis C Virus Replication, Assembly, and Host Antiviral Response"

_viruses, 2023, doi:10.3390/v15020464_

Round 1

Reviewer 1 Report

In the manuscript 'Global Lipidome profiling revealed multifaceted role of lipid species in Hepatitis C virus replication, assembly and host antiviral response', Authors performed a time kinetic study to check the enrichment of lipid droplets in the HCV life cycle, identifying different lipid species that dynamically change in relation to the time point of the viral infection. In my opinion, the topic is extremely interesting and provides data that has not been addressed in the literature. Methods and analyzes are well described. The manuscript presentation is good. 

I agree with the Authors on the opportunity to resume or complete the explanation of the study methods also in the "results" section. This helps the reader in understanding the text. The interpretation of the results obtained appears to be correct.

However, some modifications could further improve its content and/or form.

In particular:

·         In my opinion the text is excessively long. If possible, the manuscript should be summarized. The possible help of supplementary materials may be useful.

·         In light of the role of lipids in the HCV life cycle, several studies (eg: PMID: 32782961) have shown that viral eradication is associated with a significant increase in circulating cholesterol levels. The role of viral eradication should be mentioned within the discussion and would also allow to provide a clinical cut to the manuscript.

·         A summary figure on the interactions between HCV and lipids could help the reader in understanding the text.

·         The text frequently refers to growing resistance to DAAs. However, in reality this problem is marginal. Please debunk these claims.

·         Several typos and spacing errors are present in the text and should be corrected.

The Editor is invited to evaluate the appropriateness of anticipating the results of the study at the end of the introduction, as done by the Authors.

For these reasons, I would consider this study suitable for publication in “Viruses” after minor revision.

Author Response

Dear Sir, 

I have enclosed response to reviewer 1 in the attachment. 

Thank you

Reviewer 2 Report

The study by Dr Khursheed Ul Islam on the Global Lipidome Profiling Revealed Multifaceted Role of Lipid Species in Hepatitis C Virus Replication, Assembly and Host Antiviral Response, shows the accumulation of lipid droplets in the cell using confocal microscopy, and detects the changes in the lipidomic profile of the cell using mass spectrometry, at different time points (early and late) during the course of HCV infection.

Comment 1: Concerning the effect of HCV infection on the accumulation of lipid droplets, as it is already mentioned by the authors, the study confirms previously published data. Apart from the references included in the manuscript, a recent publication by Andrea Galli et al, shows that different HCV strains induce variable levels of lipid droplets accumulation and that Core localizes on lipid droplets to different extents in different HCV strains during infection in cell-culture. The authors should state what new the present study highlights about the effect of HCV infection on the accumulation of lipid droplets in the cell. 

Comment 2: The authors use immunostaining using anti-dsRNA antibody to detect HCV infection at early hours post-virus inoculation (Figure 1b), however at Figure 2, up to 3 days post-infection they provide no evidence concerning the presence of HCV in the cells which accumulate lipid droplets.

Comment 3: Important information concerning the frequency of the cell culture medium renewal is also missing. Without a change in the culture medium, after seven days of incubation, substantial energy shortage conditions resembling starvation could have been developed in the cell culture. Starvation has been shown to affect the accumulation of lipid droplets in the cell.

Comment 4: For the reason referred in comment 3, it is also essential to use mock cells/extract from mock cells, from at least two time points, one corresponding to the early hours of infection and one corresponding to the late hours of infection. The respective information concerning the incubation time of the mock cells used in lipid droplets staining in Figure 2 and in the lipidomic analysis, is missing.

Comment 5: The manuscript would benefit from analysis of alterations in the expression of genes/enzymes involved in the production and metabolism of lipids, or from detection of changes in the activity level of signaling pathways that affect the production of specific lipids, in the different time points of incubation of infected cells.

Comment 6: The manuscript would benefit from the detection of the expression of genes induced by IFN I signaling, due to HCV infection, at the different time points of incubation of cells

Comment 7: The authors should clearly state in which experiments Huh7.5 cells were used

Comment 8: Please verify that the fluorescence in the qPCR reactions was monitored at the annealing step as referred in the line 129, as different annealing and extension conditions have been used, and usually the fluorescence in the qPCR reactions is monitored at the end of the extension step.

Comment 9: It is not clear how much sample volume was injected in order to perform UPLC-ESI-MS, in the line 197 the authors write “a 5 ml of sample injection volume” whereas in the line 202 they write “1 ?l of sample injection volume “

Comment 10: Part 3.1 and Part 3.2 of results could be merged in one.

Comment 11: Several sentences need to be corrected/rephrase in order to be understood, for example:

Lines 71-73: “It has been studied to influence the lipid composition of cellular membranes, as demonstrated earlier that HCV replicates well in cholesterol and sphingolipid-rich membranes”

Lines 121-122: “ZORBAX RRHD Eclipse Plus C18, 2.1 100 mm column, 1.8 um (Agilent Technologies, USA)”

Lines 136-138: “The cell culture supernatant collected from Huh7.5 cells expressing JFH-1/GND (replication defective virus) was used as a negative control or uninfected cells (mock cell)”

Lines 241-243:” To determine the enrichment or accumulation of LDs during HCV infection, Huh7 cells were infected at various time points (2hr, 4hr, 8hr, 10hr, 12hr, day1, day3, day5 and day7) and stained with BODIPY (marker for LD) as shown earlier”

Lines 349-351: ”These lipids seem to assist in HCV replication and virion assembly while being exploited by virus during early infection thus providing an instinct that these lipids may have antiviral role”

etc

Comment 12: Give the full name of ESI the first time you use its abbreviation

Comment 13: Please provide information concerning the database used to nomenclature the identified lipids

Author Response

Dear Sir, 

I have enclosed response to reviewer 2 in the attachment. 

Thank you

Round 2

Reviewer 2 Report

I consider that all my comments have been properly answered and that the information provided was sufficient. The relevant information has been incorporated into the manuscript. I suggest only minor corrections concerning the editing of the text.

Some examples are given below:

Line 267: “HCV nonstructural proteins including NS4B has been shown..” should be converted to plural form

Line 283: “…to develop a new drug targets.”

Line 294: “…length 100mm purchased from Agilent’’ a verb is missing

Line 305: “…Retinoic acid inducible gen-I”

Line 306: “These cells were utilized for culturing HCV while as all other experiments…”

Line 314: “…as mock cells in whole of the manuscript”

Lines 315-316: “Viral supernatant having MOI of ~0.5 was used to infect naïve Huh7 cells for further experiments.” please rephrase

Lines 384-385: “Cells treated with J6/JFH-1 or JFH1/GND were incubated in their respective supernatant (JFH-1 soup/GND soup) for 4-6 hours.” please rephrase

Line 388: “Mock cells were run parallel with HCV infected cells...”

etc

Author Response

Dear reviewer, 

I have attached response to reviewer for your kind consideration. 

Thank you

Regards

Jawed
